# Subsequent Acute Ischemic Stroke in a Patient with Monocular Vision Loss Associated with Isolated Internal Carotid Artery Occlusion: A Case Report

**DOI:** 10.3390/neurolint17010003

**Published:** 2024-12-26

**Authors:** Jessica Seetge, Balázs Cséke, Zsófia Nozomi Karádi, Eszter Szalai, Valéria Gaál, László Szapáry

**Affiliations:** 1Stroke Unit, Department of Neurology, University of Pécs, 7624 Pécs, Hungary; j.seetge@gmx.de (J.S.); karadi.zsofia@pte.hu (Z.N.K.); 2Department of Emergency Medicine, University of Pécs, 7624 Pécs, Hungary; cseke.balazs@pte.hu; 3Department of Ophthalmology, University of Pécs, 7624 Pécs, Hungary; szalai.eszter@pte.hu (E.S.); gaal.valeria@pte.hu (V.G.)

**Keywords:** acute ischemic stroke, monocular vision loss, central retinal artery occlusion, internal carotid artery occlusion, intravenous thrombolysis, case report

## Abstract

**Background/Objectives:** Acute retinal ischemia, including central retinal artery occlusion (CRAO), is recognized as a stroke equivalent by the American Heart Association/American Stroke Association (AHA/ASA), necessitating immediate multidisciplinary evaluation and management. However, referral patterns among ophthalmologists remain inconsistent, and evidence-based therapeutic interventions to improve visual outcomes are currently lacking. CRAO is associated with a significantly elevated risk of subsequent acute ischemic stroke (AIS), particularly within the first week following diagnosis, yet the role of intravenous thrombolysis (IVT) in this setting remains controversial. This case report presents a unique case of CRAO with concurrent internal carotid artery (ICA) occlusion, followed by an AIS affecting the middle cerebral artery (MCA). **Case presentation:** An 83-year-old woman presented with acute, painless monocular vision loss to the emergency department. IVT was administered within 4.5 h of admission for suspected CRAO associated with ICA occlusion (ICAO) identified on CT-angiography (CTA). One hour post-thrombolysis, CT-perfusion (CTP) confirmed MCA occlusion (MCAO), necessitating mechanical thrombectomy (MT). Successful recanalization was achieved without complications, and the patient demonstrated no functional impairments at discharge. **Conclusions:** This case underscores the importance of maintaining a vigilant approach to stroke management in CRAO patients. It highlights the diagnostic challenges encountered in clinical practice and advocates for further research into the role of IVT in CRAO cases with ICAO, emphasizing the need for consensus in treatment.

## 1. Introduction

The American Heart Association/American Stroke Association (AHA/ASA) released a consensus statement in 2013, defining acute ischemic stroke (AIS) as an “episode of neurological dysfunction caused by focal cerebral, spinal or retinal infarction”. As a result, any form of acute retinal ischemia is considered a stroke equivalent, warranting immediate referral for medical evaluation [1]. However, the standard clinical approach used by most ophthalmologists differs. According to a survey of healthcare providers in Switzerland, only 64.2% of ophthalmologists would refer a patient with central retinal artery occlusion (CRAO) for urgent stroke evaluation, with only 23.9% considering systemic thrombolysis an appropriate treatment [2].

CRAO carries significance beyond visual loss, as it often indicates underlying systemic vascular disease, coming with a 44-fold increased risk of AIS within the first week following diagnosis compared to subsequent weeks [3]. Despite this, the management of CRAO remains controversial due to the lack of clear, evidence-based guidelines, especially regarding the role of intravenous thrombolysis (IVT). While IVT and mechanical thrombectomy (MT) are established treatments for AIS, their efficacy and safety in CRAO case, particularly those complicated by concurrent internal carotid artery occlusion (ICAO), remain uncertain.

This case report underscores the importance of timely recognition of CRAO as a neuroophthalmological emergency, and advocates for further research into the role of IVT in cases complicated by ICAO, highlighting the need for consensus on treatment strategies.

## 2. Case Presentation

On 2 December 2023, an 83-year-old female with a past medical history of hypertension, dyslipidemia, diabetes mellitus, and non-ST-segment-elevation myocardial infarct (NSTEMI) presented to the emergency department with sudden, right-sided vision loss. According to the patient, she was reading a book at approximately 10 a.m., after which she experienced eye fatigue. About 10–15 min later, she noticed a blurry vision in her right eye. She denied new-onset headache, myalgia, jaw claudication, diffuse posterior neck pain, scalp tenderness, nausea or vomiting, speech impairment, numbness or weakness in the extremities, facial droop, or other focal neurologic deficits. Apart from wearing reading glasses (left eye +4.0D), she had no significant ophthalmic history. No history of tobacco use or alcohol consumption was reported. Her current medications included perindopril, bisoprolol, aspirin, atorvastatin, trimetazidine, metformin, indapamide, carbamazepine, and pantoprazole. There were no reported allergies to medications. Upon arrival, she was triaged as a potential acute stroke patient, and both the neurology and ophthalmology departments were alerted.

On initial assessment in the emergency department, vital signs, including blood pressure (147/73 mmHg), heart rate (78/min), and blood glucose (6.87 mmol/L) were recorded. The platelet count, coagulation parameters (prothrombin time, international normalized ratio, and activated partial thromboplastin time), and C-reactive protein (CRP) levels were within normal limits. The neurological examination was unremarkable, except for visual impairment.

CT/CT-angiography (CTA) performed to assess carotid disease revealed no evidence of intracranial hemorrhage or acute ischemia (manual Alberta Stroke Program Early CT Score [mASPECTS] 10, Figure 1) but indicated an occlusion of the right internal carotid artery (ICA).

A bedside ophthalmological evaluation, including a dilated funduscopic examination, did not show signs of central embolization or ischemia (edematous neuroretina, macular “cherry-red” spot, or attenuated arteries with slow segmental blood flow). However, a right-sided relative afferent pupillary defect (RAPD) was revealed. Visual acuity in the right eye was measured at 0.2 (40/200), with an inferior visual field defect. Anterior segment examination and intraocular pressure measurements were normal. The left eye examination was unremarkable, with a measured visus of 0.4 (40/100).

In the absence of contraindications, IVT with 40 mg alteplase (administered at a rate of 0.9 mg/kg with 10% given over 1 min and the remainder over 59 min) was initiated within a 4.5 h time window. The decision to proceed with IVT was made following a comprehensive risk–benefit assessment conducted by the multidisciplinary team, considering the patient’s age, overall clinical stability, and underlying comorbidities. Although the risk of bleeding complications was acknowledged, the potential benefits of restoring perfusion and improving ophthalmological outcomes justified the treatment decision. Alternative options, such as standard care, were considered but deemed insufficient given the acute vision loss.

Adequate flow via the anterior communicating artery (ACA) resulted in perfusion of the right hemisphere from the left side without any signs of apparent hemispheric deficits. Thus, neurointervention was not pursued initially.

While there was a slight improvement in the patient’s right-sided vision following IVT, one hour later, she experienced a sudden onset of left-sided central facial and hemiparesis accompanied by severe dysarthria. This acute neurological deterioration prompted immediate reassessment, and a subsequent CT-perfusion (CTP) scan revealed a large vessel occlusion (LVO) of the middle cerebral artery (M1 segment), with favorable imaging parameters: mASPECTS 9, a core volume of 15 mL, and a mismatch ratio of 2.6. These findings, combined with the abrupt worsening of symptoms and the significant penumbra of salvageable tissue (39 mL), supported the decision to proceed with MT (Figure 2). Conservative treatment at this stage was considered inappropriate due to the risk of further neurological decline.

The intervention was performed through puncture of the right femoral artery, leading to successful thrombus extraction and complete recanalization (treatment in cerebral ischemia [TICI] 3, Figure 3). No complications were observed during the intervention.

After MT, the patient was transferred to the neurology department. She showed a significant recovery with only mild left-sided hemiparesis and dysarthria. There was a noticeable improvement in her right eye’s vision, allowing her to read digits from a distance of 1 m confidently. Follow-up CT scans showed no signs of hemorrhagic or new ischemic lesions. Routine laboratory tests revealed HbA1c and lipid levels within the target ranges.

Upon hospitalization, a transthoracic echocardiogram (TTE) showed concentric left ventricular hypertrophy and inferoseptal hypokinesia despite adequate left ventricular function. During monitoring, atrial fibrillation was detected, prompting the initiation of low-dose direct oral anticoagulant (DOAC) therapy due to a high CHA_2_DS_2_-VA score, in addition to statin therapy.

On the 7 December, during a follow-up ophthalmologic examination, hyperreflectivity of the inner nuclear layer (INL) consistent with paracentral acute middle maculopathy (PAMM), was noted in the right eye, suggestive of a possible microembolization. Vision impairment in the left eye was observed and attributed to traction caused by the vitreous membrane (vitreomacular adhesion [VMA] and vitreomacular traction [VMT]). Visual acuity of 0.3 (40/120) was assessed bilaterally.

The clinical presentation is consistent with a right CRAO, likely resulting from right ICAO, with suspected cardiogenic embolization as the underlying cause. The patient was discharged with no functional impairment (Figure 4).

## 3. Discussion

Acute retinal ischemia is a rare neuro-ophthalmologic emergency, with an incidence of 1.9 per 100.000 person-years in the United States [4]. It is characterized by sudden, painless, monocular vision loss, most frequently caused by CRAO [5]. Nonarteritic cases, accounting for 95% of CRAOs, typically result from an embolus or thrombus originating from the internal carotid artery or heart, whereas arteritic cases, comprising 5% of instances, are commonly associated with giant cell arteritis (GCA) [6].

CRAO predominantly affects both central vision (visual acuity) and peripheral vision (visual fields), along with color- and stereovision. Although retinal artery occlusions are defined based on a compelling clinical history (acute-onset, painless monocular vision loss), the diagnosis is often reinforced by an RAPD and characteristic funduscopic findings indicative of retinal hypoperfusion [6]. Acute CRAO classically presents with retinal edema (whitening), a cherry-red spot in the fovea (preserved choroidal circulation of the macula), and attenuated arteries showing stagnant segmental blood flow, described as “box-carring” upon fundoscopic examination, but may be absent or subtle [7]. The optic nerve head retains perfusion despite occlusion due to an alternative blood supply from the short posterior ciliary arteries. Further diagnostic assessments such as optical coherence tomography (OCT)-based determination of inner retinal layer hyperreflectivity may be considered to distinguish between incomplete and complete CRAO regarding possible outcome [8]; however, treatment should not be delayed as a result [6]. The ASA/ASH recommends immediate cerebral imaging for all suspected cases of retinal ischemia. Consequently, interdisciplinary collaboration is essential to anticipate potential cerebral ischemic events in patients with CRAO [9].

The risk of CRAO increases with age, particularly in individuals > 80 years, with an incidence of 10.1 per 100,000 person-years [10]. Additionally, individuals with cardiovascular risk factors such as hypertension, hypercholesterolemia, diabetes mellitus, tobacco use, and obesity are at increased risk [11]. Several studies have demonstrated an elevated risk of inpatient stroke, acute myocardial infarction, and death in individuals with a history of retinal artery occlusion, with a combined risk of up to 19% [12]. A systematic review conducted in 2020 revealed that 30% of patients with acute CRAO showed signs of acute cerebral ischemia on magnetic resonance imaging (MRI) performed within seven days, highlighting the need for admission to a stroke center for close monitoring and etiological evaluation [13,14]. Another study indicated a 2.2% risk of symptomatic ischemic stroke within 15 days preceding and following CRAO [15]. A prospective study involving 77 CRAO patients as part of the European Assessment Group for Lysis in the Eye (EAGLE) study reported that within one month, 6% of participants were diagnosed with a new-onset stroke [11]. Additionally, CRAO patients have a 2.7-fold greater lifetime risk of developing a stroke compared to controls [16]. Predictors associated with inpatient stroke include female sex, smoking status, alcohol dependence, hypertension, and carotid artery stenosis [12]. A single-center study involving 103 CRAO patients revealed that 37% of patients exhibited ipsilateral critical carotid stenosis during CT/MRI-angiography [17], although complete occlusion, as seen in our case, is only rarely reported. In the EAGLE study, carotid artery stenosis of ≥70% was detected in 40% of all enrolled patients during Doppler ultrasonography, compared to 10–30% of patients with cardiovascular risk factors alone and 4.1% of the general population [18]. Our patient exhibited all mentioned risk factors associated with inpatient stroke, except for a history involving smoking and alcohol consumption.

To enhance public awareness, adding a “B” for balance and an “E” for eyes to the FAST (Face, Arm, Speech, Time) acronym, resulting in “BE-FAST,” has been proposed to improve the recognition of the condition [19]. While recent studies have associated the administration of IV alteplase within a 4.5-h window from symptom onset with a higher likelihood of significant visual recovery [20], visual outcomes in CRAO patients are generally poor, with only 50% of treated patients achieving functional visual acuity in the affected eye [21], compared to 17% of patients with a natural progression [6]. Tobalem and colleagues suggest that ganglion cell infarction may occur within as little as 15 min of blood flow interruption, leaving a narrow window for effective intervention [22]. However, patients with CRAO are often offered treatment even if symptoms persist beyond 4.5 h, mainly when uncertainty exists regarding the completeness of occlusion [22]. Consequently, management of acute CRAO patients primarily focuses on secondary prevention measures, including monitoring for complications and reducing the risk of subsequent ischemic events such as cerebral ischemia, myocardial infarction, and cardiovascular death [6].

A study conducted in 2020 found that 30% of patients experiencing minor strokes due to internal carotid artery occlusion (ICAO) experienced early neurological deterioration (END) following IVT, likely attributed to clot fragmentation and subsequent distal embolization [23]. Similarly, patients who underwent carotid endarterectomy (CEA) or carotid artery stenting (CAS) for extracranial ICAO after administration of IVT were at higher risk of post-procedure strokes compared to those without prior IVT treatment [24]. Conversely, results from the Interventional Management of Stroke III (IMS III) trial indicated that 81% of patients with ICAO treated with MT achieved partial or complete recanalization within 24 h, compared to only 35% of IVT patients [25]. In our case, administration of IVT could have influenced the risk of subsequent AIS by potentially increasing the probability of thrombus migration, a phenomenon previously seen in patients with cerebral ischemic stroke treated with alteplase [26,27]. Since there is no indication for MT in cases of isolated ICAO, this raises the question of whether refraining from recanalization therapy might be a preferable approach over IVT for CRAO patients with concurrent ICAO, particularly concerning the risk of subsequent ischemic events.

## 4. Study Limitations

This case report has several limitations. As a single-patient study, the findings are inherently limited in generalizability and cannot establish causal relationships, particularly regarding the role of IVT in the subsequent AIS. The absence of a control group restricts the ability to compare treatment outcomes or assess whether the observed ophthalmological and neurological improvements can be directly attributed to IVT or were influenced by the natural disease progression. Additionally, the presence of multiple cardiovascular comorbidities, including hypertension, diabetes mellitus, a history of NSTEMI, left ventricular hypertrophy and atrial fibrillation may have served as confounding factors that influenced both the clinical course and treatment response.

## 5. Conclusions

Any form of acute retinal ischemia is considered on par with cerebral ischemic stroke and requires immediate referral to a stroke center. To date, no widely accepted management algorithm exists, and treatment approaches vary based on local protocols and the experience of physicians. Therefore, future research efforts should focus on prioritizing the early identification of patients with acute vision loss to determine whether CRAO is the underlying cause, and thus an immediate neurovascular workup is needed. Likewise, there is a critical need to further investigate the role of IVT in patients with CRAO due to ICAO, emphasizing the importance of establishing guideline-endorsed treatment protocols.

## Figures and Tables

**Figure 1 neurolint-17-00003-f001:**
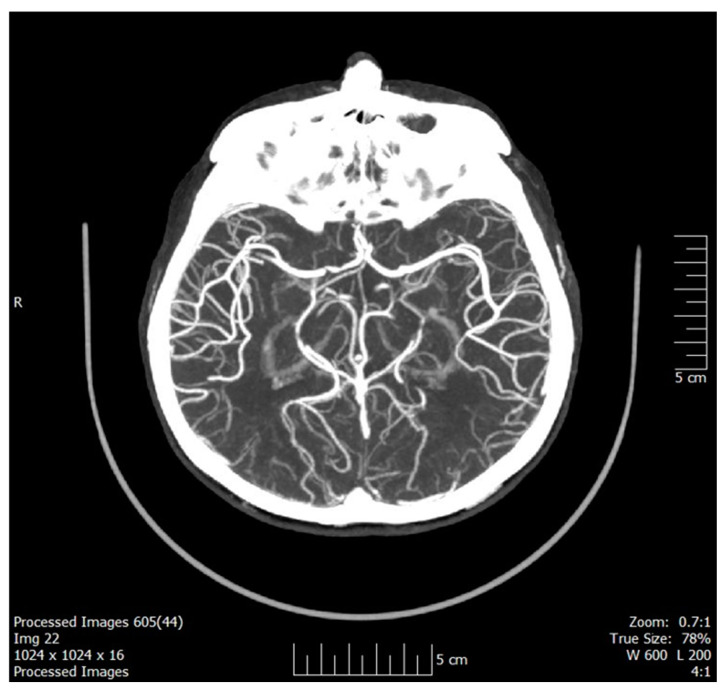
CTA before IVT showing intact cerebral large vessels. Abbreviations: CTA = CT-angiography; IVT = intravenous thrombolysis.

**Figure 2 neurolint-17-00003-f002:**
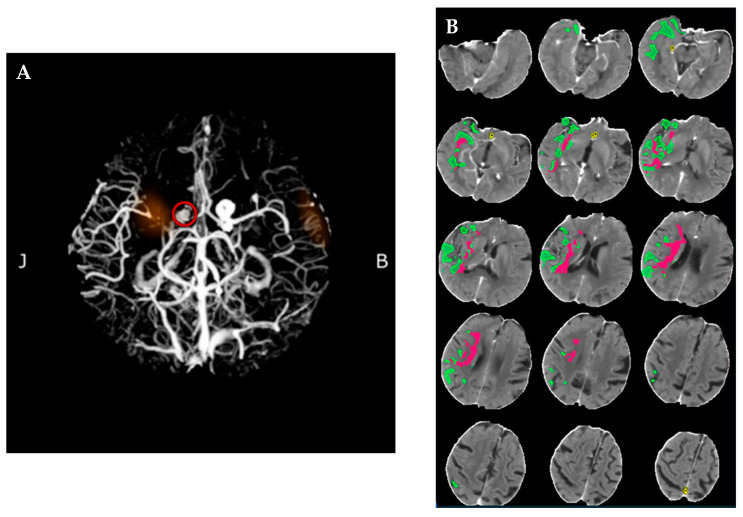
Panel (**A**) shows the CTA image identifying a LVO at the M1 segment of the MCA (circle in red). Panel (**B**) displays the CTP images acquired during the second CT examination, illustrating perfusion deficits with areas of reduced cerebral blood flow (highlighted in pink) and areas of preserved or delayed perfusion (highlighted in green). Abbreviations: CTA = CT-angiography; LVO = large vessel occlusion; MCA = middle cerebral artery; CTP = CT-perfusion.

**Figure 3 neurolint-17-00003-f003:**
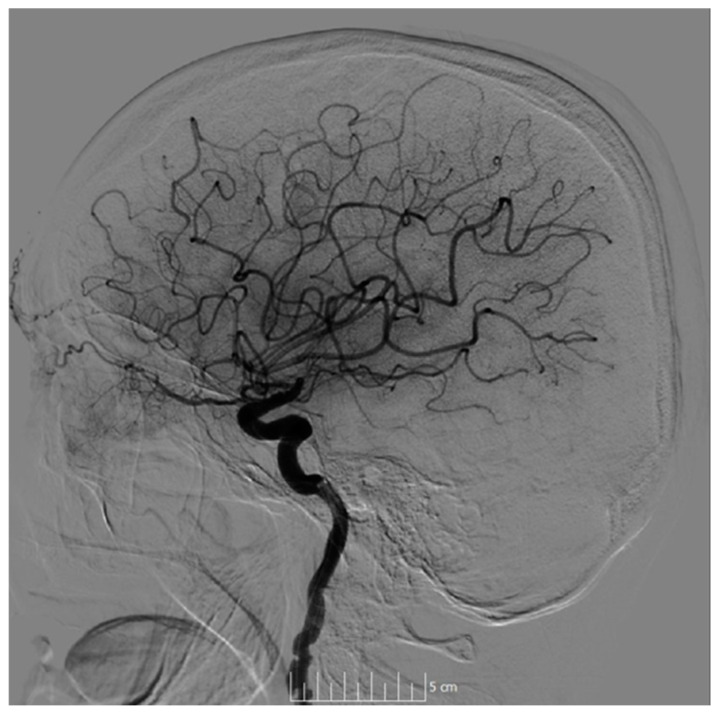
DSA showing successful recanalization (TICI 3). Abbreviations: DSA = Digital subtraction angiography; TICI = treatment in cerebral ischemia.

**Figure 4 neurolint-17-00003-f004:**
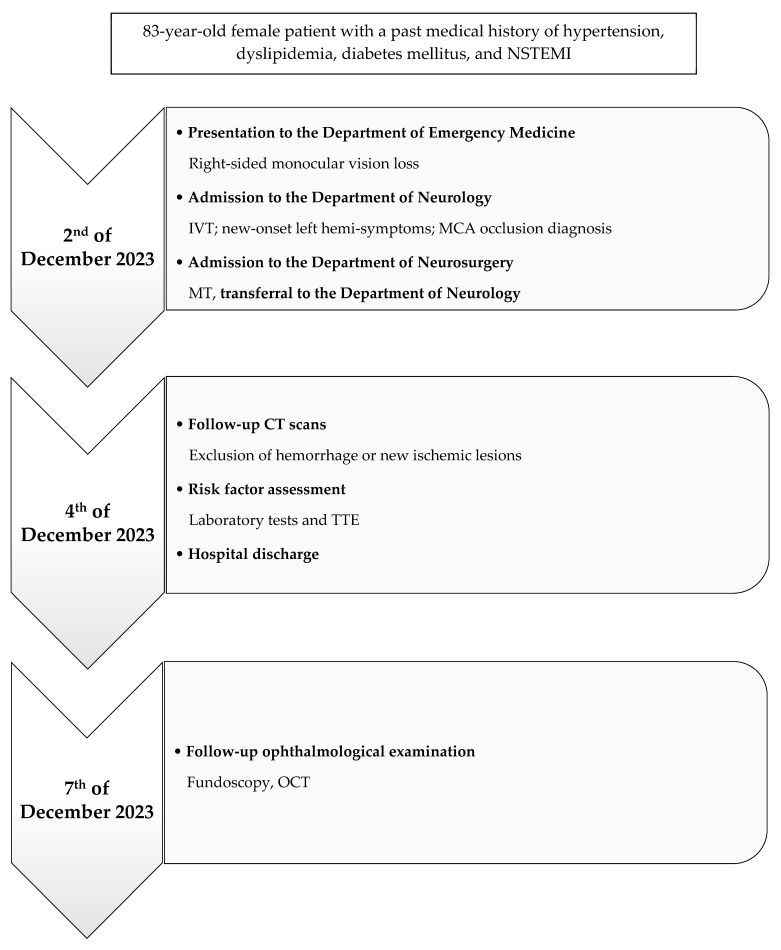
Case Report Timeline. Abbreviations: NSTEMI = non-ST-segment-elevation myocardial infarct; IVT = intravenous thrombolysis; MCA = middle cerebral artery; MT = mechanical thrombectomy; TTE = transthoracic echocardiogram; OCT = optical coherence tomography.

## Data Availability

The original contributions presented in the study are included in the article and further inquiries can be directed to the corresponding author.

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
