# Peer review of "Subsequent Acute Ischemic Stroke in a Patient with Monocular Vision Loss Associated with Isolated Internal Carotid Artery Occlusion: A Case Report"

_2035-8377, 2024, doi:10.3390/neurolint17010003_

Round 1

Reviewer 1 Report

Comments and Suggestions for Authors

You should include BOLD background/objectives, case presentation, and conclusion in the abstract.

The introduction part needs to be improved. The introduction cannot be just one sentence.

How long after the symptoms were the ambulance notified? How long did the patient get to the hospital after the symptoms?

The patient weighed only 44 kg and was administered only 40 mg of alteplase???

The discussion part needs to be improved, I suggest the following article:

https://doi.org/10.3390/jpm14060596

The study limitation part is missing

Reviewer 2 Report

Comments and Suggestions for Authors

This case report presents in detail the diagnosis and treatment process of an 83-year-old female patient. The patient presented with acute painless monocular vision loss and was found to have right internal carotid artery (ICA) occlusion upon examination. She was diagnosed with suspected central retinal artery occlusion (CRAO) and received intravenous thrombolysis (IVT) within 4.5 hours of admission. One hour after thrombolysis, the patient developed new neurological deficits, and CT perfusion (CTP) examination indicated middle cerebral artery (MCA) occlusion. Subsequently, mechanical thrombectomy (MT) was performed and was successful. The article discusses the relationship between CRAO and acute ischemic stroke (AIS), treatment methods, and related risk factors, emphasizing the importance of maintaining vigilance in stroke management in CRAO patients. Before publication, the author should address following issues:

1. The introduction is too simple. Please provide more context and background information to help readers understand the significance of your research.

2.The methodology section is unclear and lacks details about the experimental design and procedures. Please provide a more detailed description of how the data was collected and analyzed.

3.The relationship between carotid artery stenosis and CRAO and stroke was mentioned, and more detailed vascular assessment information should be provided here to better support the discussion.

4.Although the article reported the patient's condition at discharge, it did not mention the long-term follow-up results, such as the stability of visual acuity recovery, the long-term improvement of neurological deficits, and whether there was a recurrence of stroke or other cardiovascular events. Long-term follow-up data is crucial for evaluating the treatment effect and patient prognosis, and it is recommended to supplement it.

5.When deciding on IVT and MT treatments, in addition to considering the time window and imaging examination results, other factors such as the patient's overall condition and potential risk-benefit assessment should be more detailedly expounded.

6.The labeling of CTA and CTP images in Figure 2 is not clear enough. 

7.Add a detailed discussion of the treatment decision-making process in the treatment part, including how to comprehensively consider factors such as the patient's age, underlying diseases, and disease severity to weigh the risks and benefits of IVT and MT treatments, and whether other alternative treatment options have been considered, making the treatment decision-making process more transparent and reasonable.

  •  

Round 2

Reviewer 1 Report

Comments and Suggestions for Authors

The article can be published.

Reviewer 2 Report

Comments and Suggestions for Authors

The author has addressed my issues.